# Contrasting segregation patterns among endogenous retroviruses across the koala population
Mette Lillie [1,2] ✉, Mats Pettersson [1] & Patric Jern [1] ✉

Koalas (*Phascolarctos cinereus*) have experienced a history of retroviral epidemics leaving their trace as heritable endogenous retroviruses (ERVs) in their genomes. A recently identified ERV lineage, named *phaCin-β*, shows a pattern of recent, possibly current, activity with high insertional polymorphism in the population. Here, we investigate geographic patterns of three focal ERV lineages of increasing estimated ages, from the koala retrovirus (KoRV) to *phaCin-β* and to *phaCin-β-like*, using the whole-genome sequencing of 430 koalas from the Koala Genome Survey. Thousands of ERV loci were found across the population, with contrasting patterns of polymorphism. Northern individuals had thousands of KoRV integrations and hundreds of *phaCin-β* ERVs. In contrast, southern individuals had higher *phaCin-β* frequencies, possibly reflecting more recent activity and a founder effect. Overall, our findings suggest high ERV burden in koalas, reflecting historic retrovirus-host interactions. Importantly, the ERV catalogue supplies improved markers for conservation genetics in this endangered species.

Retroviruses are a diverse group of RNA viruses that require conversion of their RNA genome to DNA, which integrates irreversibly as a provirus into the host's nuclear DNA. Infection normally occurs in somatic cells, where the provirus shares the cell's fate and will thus not persist over time. If the germline is infected, however, the provirus can be transmitted to offspring as a heritable endogenous retrovirus (ERV). Over the evolutionary time-scale, ERVs have accumulated in contemporary genomes and offer a glimpse into the past retrovirus infection history[1,2].

Koalas (*Phascolarctos cinereus*), an iconic feature of Australian wildlife, have attracted considerable attention. In addition to their cultural significance and conservation concerns, they are also hosts to the koala retrovirus, KoRV[3], which occurs across a large range of the eastern koala distribution[4,5]. KoRV has been associated with wide-ranging disease outcomes, including cancers and opportunistic infections[6], and is also observed in the koala germline as endogenous KoRV (enKoRV)[7-10].

We recently reported that, in addition to the gammaretroviral enKoRV, the koala germline also contains recent accumulation of a betaretroviral ERV lineage, named *phaCin-β*, which began infiltrating the koala genome between 0.7 and 1.5 million years ago[11]. Here, we investigate the insertional polymorphism and geographical distribution of enKoRV and *phaCin-β*, and compare these to an older related ERV lineage named *phaCin-β-like*, which began colonizing the koala genome

between 2.4 and 5.5 million years ago[11]. It is relevant to evaluate the population-wide distribution of these lineages for insights into their infection history, establishment as ERVs and impact on the host population. This is pertinent given the overlapping time intervals of KoRV and *phaCin-β* infections and their potentially continuing activity in the koala population[11].

For these studies, we take advantage of the Koala Genome Survey[12], which has produced unassembled short-read whole-genome sequences from 430 individual koalas across wild populations in Queensland (QLD; $n = 100$), New South Wales (NSW; $n = 246$), and Victoria (VIC; $n = 72$) (Fig. 1a), as well as two captive populations ($n = 12$; Taronga Zoo, NSW; Featherdale Sydney Wildlife Park, NSW). We find contrasting polymorphism patterns for the recent ERV lineages across the koala population and, in total, reveal thousands of ERVs, suggesting high ERV burden in koala, especially regarding the two, possibly still active, *phaCin-β* and KoRV lineages. These ERVs reflect historic retrovirus-host interactions and supply markers for improved population conservation genetics.

## Results

### ERV polymorphism

Following our previously described ERV mapping strategy[11,13] and utilizing reference sequences derived from the three focal ERV

[1]Science for Life Laboratory, Department of Medical Biochemistry and Microbiology, Uppsala University, SE-751 23, Uppsala, Sweden. [2]Department of Ecology and Genetics, Animal Ecology, Uppsala University, SE-752 36, Uppsala, Sweden. ✉e-mail: Mette.Lillie@ebc.uu.se; Patric.Jern@imbim.uu.se

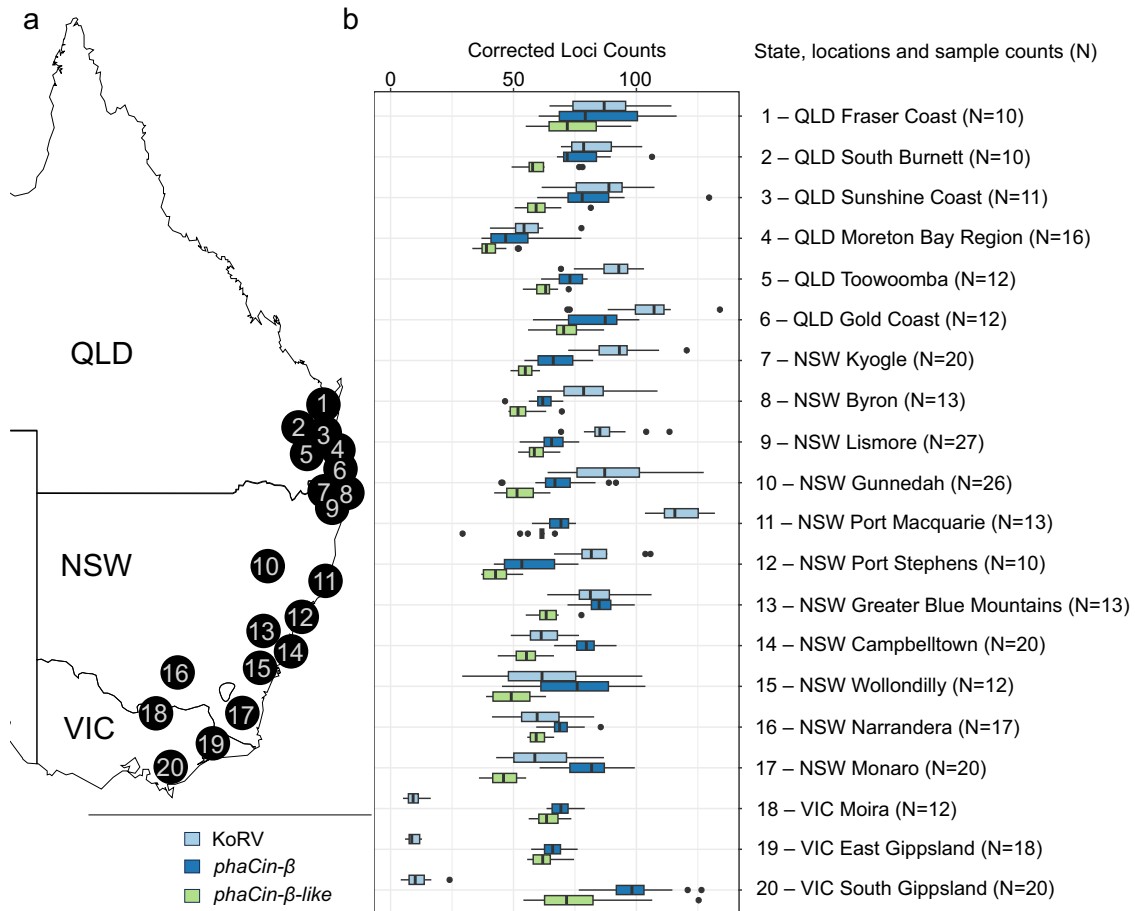

**Fig. 1 | Contrasting ERVs accumulation pattern of across the koala population.**
**a** Map of Australian east coast showing koala sampling locations (modified from
Hogg et al.[12]). **b** Boxplots of corrected ERV counts per individual across the sampling
locations with >10 samples (312 individuals from 20 locations; see details in Sup-
plementary Table 1) for KoRV (light blue), *phaCin-β* (dark blue) and *phaCin-β-like*
(green) ERVs. Locations are ordered by latitude.

lineages KoRV, *phaCin-β*, and *phaCin-β-like*, we analyzed whole-
genome short-read sequences from the koala population (Fig. 1a),
and identified a total of 12,990 ERV loci. We identified more unique
ERV loci as characterized by unique ERV-host junctions in the
younger lineages, such that 9346 loci were classified as KoRV, 3175 as
*phaCin-β* and only 469 as *phaCin-β-like* (Table 1). The timing of
retroviral activity was also reflected in the frequency distribution of
ERV insertional polymorphisms across the population. The number
of individuals having a given KoRV locus varied from 1 to 163 (mean
frequency 0.7%; median frequency 0.5%). 4642 (49.7%) KoRV loci
were private, i.e. only found in one individual koala. Individuals had
between 2 and 138 KoRV loci in the genome. Population counts per
*phaCin-β* locus varied from 1 to 430 (mean frequency 2.2%; median
frequency 0.7%). Individuals carried 38–110 *phaCin-β* loci each, and

1015 (32.0%) *phaCin-β* loci were private to a single individual. In
contrast, *phaCin-β-like* loci showed higher frequencies (mean fre-
quency 12.9%; median frequency 1.4%), reflecting their earlier
expansion history, and individuals contained between 27 and 73
*phaCin-β-like* integrations in their genome. There were still relatively
many rare *phaCin-β-like* loci, with 151 (32.2%) *phaCin-β-like* inser-
tional polymorphisms found in only one individual.

The identified ERV loci were widely and unevenly distributed across
the genome (Supplementary Fig. 1), with indications of integration "hot-
spots" regions emerging from population wide identifications, for example,
a 20 kb region MSTS01000019.1:3,020,001-3,040,000 that contained 5
*phaCin-β* and 5 KoRV insertions (Supplementary Fig. 1). This particular
region overlaps with a predicted LRP1B gene (MSTS01000019.1:1,511,885-
3,684,244), a member of the low-density lipoprotein (LDL) receptor family
involved in various normal cell functions and development, and whose
disruption is associated with several types of cancer[14]. In some cases,
*phaCin-β* and KoRV integrations are particularly close, for example, at
MSTS01000042.1:9,858,795-9,858,778 there is only 17 bp separating inte-
grations by *phaCin-β* and KoRV in different samples (Supplementary
Fig. 2). Assuming a neutral model of random integrations where the success
rate equals the number of unique ERVs divided by the genome length, we
would not expect to find any 20 kb window with 5 or more ERVs (exact
binomial test $p \approx 8.3 * 10^{-8}$). However, we observe 166 such windows with 5
or more integrations. We also observe potential "coldspot" regions of the
genome, such as the 3.7 Mb region MSTS01000019.1:5,903,013–9,671,599
where no integrations by the three focal ERV lineages KoRV, *phaCin-β*, and
*phaCin-β-like* were detected in our population-wide screening. Assuming

**Table 1 | ERV polymorphism across koala samples (*n* = 430)**

| ERV | Total ERVs[a] | Private ERVs | Frequency | | | ERVs per koala |
|---|---|---|---|---|---|---|
| | | | Max | Mean | Median | |
| KoRV | 9346 | 4642 (49.7%) | 38% | 0.7% | 0.5% | 2–138 |
| *phaCin-β* | 3175 | 1015 (32.0%) | 100% | 2.2% | 0.7% | 38–110 |
| *phaCin-β-like* | 469 | 151 (32.2%) | 100% | 12.0% | 1.4% | 27–73 |

a. Number of identified locations containing an ERV in at least one individual.

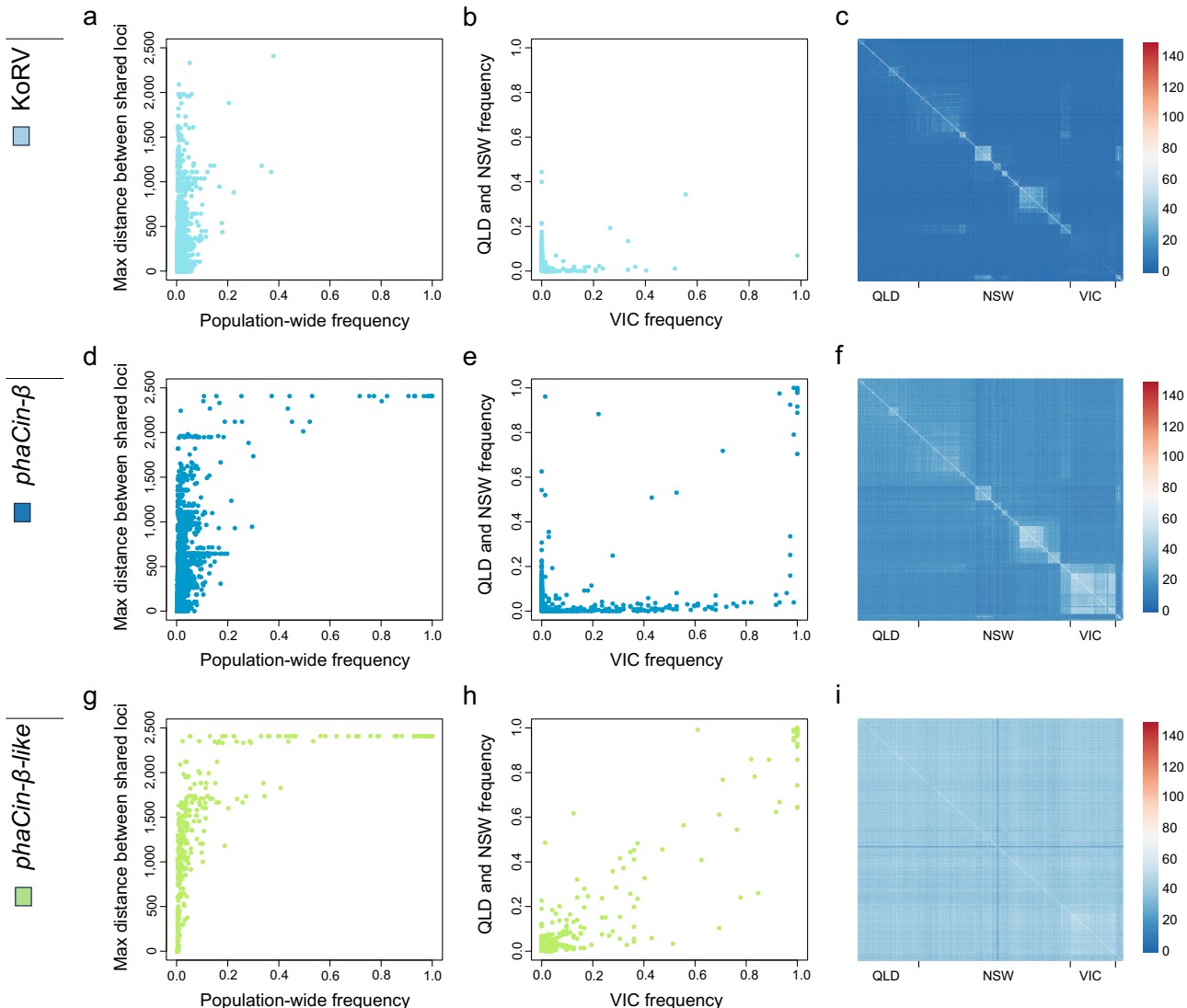

**Fig. 2 | Contrasting geographic ERV frequencies among koalas.** Comparisons are shown for the three focal KoRV (**a–c**), *phaCin-β* (**d–f**), and *phaCin-β-like* (**g–i**) ERV lineages across the koala population. **a, d, g** Maximum geographical (haversine) distance between individuals sharing an ERV integration versus the population-wide frequency for KoRV (**a**), *phaCin-β* (**d**), and *phaCin-β-like* loci (**g**).

**b, e, h** Integration frequency in Queensland (QLD) and New South Wales (NSW) versus frequency in Victoria (VIC) for KoRV (**b**), *phaCin-β* (**e**), and *phaCin-β-like* loci (**h**). **c, f, i** Heatmaps of pairwise shared ERV counts between individuals for KoRV (**c**), *phaCin-β* (**f**), and *phaCin-β-like* loci (**i**). Individuals are ordered in sample regions approximately by latitude.

the same neutral model (above), the probability of observing a gap of this length can be estimated to $p \approx 2.9 * 10^{-7}$.

## Geographical distribution of ERV polymorphism

Relative numbers of KoRV, *phaCin-β* and *phaCin-β-like* loci varied across the sampling distribution, most noticeably the difference in KoRV counts between QLD/NSW and VIC (Fig. 1b), reflecting well documented differences in disease prevalence and KoRV distribution[4,10,15]. Local variation within northern populations is also apparent, such as Port Macquarie samples having comparatively high KoRV counts per individual. In QLD and NSW, the number of KoRV integrations per individual are either comparable or greater than the number of *phaCin-β* integrations. In VIC, the greatest number of integrations per individual were *phaCin-β*, then *phaCin-β-like*, with very few KoRV.

Different patterns of polymorphism and per-locus frequency were observed across the koala distribution (Fig. 2). Frequency comparisons (Fig. 2b, e, h) for NSW/QLD vs VIC for the older ERV lineage, *phaCin-β-like* (Fig. 2h), show strong positive correlation ($r^2 = 89.7\%$), such that integrations at a higher frequency in NSW and QLD are also at higher frequency in

VIC. This pattern is largely absent from *phaCin-β* ($r^2 = 26.3\%$) and KoRV ($r^2 = 2.1\%$) (Fig. 2b, e), where intermediate-to-high frequency loci in NSW/QLD are at low frequency, or not present, in VIC, and vice versa. These patterns provide strong evidence that the majority of loci in the two younger lineages entered germline after the current koala populations were established, while *phaCin-β-like* loci must pre-date this establishment. Interestingly, there is a small number of *phaCin-β* loci with similar, and high frequencies in both groups, indicating that the infection-window had opened up before the populations fully separated. Pairwise counts of shared integrations (Fig. 2c, f, i) show that there is generally little sharing across wider regions in QLD and NSW regarding *phaCin-β* and KoRV, although some localized regions show greater locus sharing of both ERV lineages. These regions with greater ERV sharing are likely the result of inbreeding in small, insular populations. There is a large degree of *phaCin-β* locus sharing across VIC with intermediate frequencies of many *phaCin-β* ERV loci private to the state, which may reflect founder effect (Supplementary Fig. 3). The *phaCin-β-like* integrations show no apparent pairwise structure across the koala population, consistent with a longer residence in the koala germline.

## Comparison of *phaCin*-β insertion polymorphism in koala genomes assemblies

Recently, a second high-quality koala genome assembly became available[16]. The newly sequenced individual was from the bottlenecked South Australian (SA) population, allowing us to compare its *phaCin-β* ERV integrations to those of the reference koala assembly ("Bilbo"), which was from QLD. Using RetroTector[17] and BLAT[18], we identified 45 *phaCin-β* integration sites in Bilbo, out of which 17 are solo LTRs and 28 proviruses, ranging between 6377 and 8946 bp (median: 8433 bp). In comparison, the SA genome contains 81 *phaCin-β* integration sites, including 17 solo LTRs and 66 proviruses ranging between 5638 and 8518 bp (median: 8464 bp). A total of 14 integration sites with proviruses and 13 solo LTRs were shared between the two reference genomes (Supplementary Table 2). Of the 28 provirus loci in Bilbo, two (7%) were solo LTRs and 12 (43%) were empty pre-integration sites in SA (Supplementary Table 2). Of the 66 provirus loci in the SA assembly, two were solo LTRs (3%) and 43 (65%) were empty pre-integration sites in Bilbo (Supplementary Table 2).

## Discussion

Considering the recent listing of the northern koala populations as an endangered species by the Australian Government[19], it is relevant to improve the population conservation genetics and to better understand the potential threats presented by deleterious genetic elements. Here, we leverage our ERV screening approach[11,13] across a unique koala population genomics dataset[12] to compare the polymorphism and distribution pattern of three recent ERV lineages, the gammaretroviral KoRV and the two betaretroviral *phaCin-β* and *phaCin-β-like*[11]. The main outcomes from studying these three ERV lineages in population-wide screening are two-fold. First, the observed frequency differences of the three focal ERV lineages across the koala population provide increased resolution over conventional de novo single nucleotide polymorphisms (SNPs), due to the abundance of loci segregating at divergent frequencies in different koala populations. This supports accurate tracking of koalas from the different subpopulations for improved conservation genetics. Second, contrasting frequency patterns inform about retroviral infection history of KoRV, which is most common in northern koalas and *phaCin-β* found at high frequencies in the south, compared to *phaCin-β-like*, which is a young lineage compared to other ERVs but much older than both KoRV and *phaCin-β*[11], showing more uniform frequency across the koala population.

We detect thousands of ERV integrations across the koala genome, and decreasing abundance with predicted ERV lineage ages. KoRV, which is the youngest lineage in the study, is the most prolific ($n = 9346$) with high number of loci private to individual koalas (49.7%), reflecting the present KoRV activity in the koala population[3]. Consequently, many of these ERVs are likely to be lost from the population within a few generations. There are fewer than half as many loci of the slightly older *phaCin-β* ($n = 3175$) lineage, with lower number of private loci (32%). This lower number, relative to KoRV, could be partly the result of many ERVs being lost due to drift since *phaCin-β* entered the koala population or time since the potential extinction of *phaCin-β* (although we cannot exclude *phaCin-β* from still being active). Alternatively, given the overlapping age estimates between KoRV and *phaCin-β*[11], differences in numbers of identified loci could also reflect varying degree of activity in the koala population and, by extension, the amount of ERVs that were inherited through the germline. Another potential explanation for these patterns could be differences in *phaCin-β* and KoRV integration efficiency. Since *phaCin-β* is estimated to have entered the koala before KoRV, but have fewer ERVs with population-wide spread, it is conceivable that infections were less permissive to ERV establishment or possibly suppressed in the koala. A host of mechanisms against retroviral replication have been described[1,2], but remain uncharacterized in koala. Such an innate defense mechanism in koala could involve Piwi-interacting RNAs (piRNAs), which normally suppress transposition in trans by means of anti-sense transcripts but appears to inhibit KoRV replication *in cis* by sense strand piRNA from unspliced KoRV transcripts that halt

replication[20]. Along these lines, *phaCin-β-like* being the oldest of the three focal ERV lineages identified only a few hundred ($n = 469$) loci.

The distribution across the genome of these three ERV lineages was widespread and uneven, showing apparent integration "hotspots". In particular, hotspot regions sharing many KoRV and *phaCin-β* integrations indicate that specific regions of the genome have been accessible for hundreds of thousands of years. On the other end, there are apparent "cold-spots" in the koala genome, where no integrations by our focal three ERV lineages are detected across the koala population. An explanation for these observations could be selection against deleterious insertions in particular regions of the genome, or that some retroviruses have been observed to exhibit integration preference as a result of interactions between their viral structural proteins with host chromatin and cellular proteins[21]. For example, the lentiviral HIV-1 displays targeted integrations within gene-dense genomic regions, predominantly within active transcription units[22,23]. Murine leukemia virus (MLV) and other studied gammaretroviruses similarly target gene-dense regions, often in the immediate vicinity of promoters and CpG islands[24–26]. Thus, the observed integration patterns in the koala likely reflect the chromatin status of the genome, with accumulation of integrations in accessible and acceptable regions of the genome and absence in regions that are not. Regions of high integration instances from the three ERV lineages may also indicate regions of the genome that have been consistently accessible to retrovirus integration over millions of years.

The widely recognized, major genetic differentiation between northern (NSW and QLD) and southern (VIC and SA) koala populations has directed most population management in the past[15,16,27]. The main contributor to this differentiation is the effective extinction and subsequent reintroduction of koalas into the southern states from relict island populations during the early 20th century[16]. As a result, the southern states have very low genetic diversity on the SNP level[27–29]. We observed this genetic divide between the northern and southern states in our data, with a marked difference in KoRV integration counts across the regions, reflecting the geographical differences in KoRV prevalence[5]. We also observed differences in the pattern of *phaCin-β* integrations across the states. Given the stationary lifestyle of koalas, these patterns could reflect the historical intensity of infection, which likely differ between the two retrovirus lineages. Comparisons between the provirus content of the SA genome relative to the northern Bilbo genome would imply more recent *phaCin-β* activity in the southern koala populations. The patterns in VIC, though, appear to be shaped by a founder effect (Supplementary Fig. 3). VIC has prominent locus sharing across the whole sampled region, in contrast to the patterns in QLD and NSW, which could be the result of a founder effect, after populations were reintroduced from island refugia. The relative contributions from any founder effect on ERV loci, compared to the recent, potentially ongoing, *phaCin-β* activity in these southern population will require further study. In particular, investigating the proviral content and sequence divergence of the ERV integrations in VIC populations, for example using long-reads, will greatly inform our understanding of the two processes.

Despite the koala populations of the southern states having low genetic diversity[27–29], we observe considerable insertional diversity by *phaCin-β*, with a substantial number of loci segregating close to 50% allele frequency. While ERV polymorphism can be valuable in tracing the co-evolutionary history of host species and retroviruses, ERVs can also be viewed as informative genetic markers, especially in cases of species where relatively recent retroviral activity overlaps with the establishment of present-day population structure. ERVs could be particularly informative genetic markers in conservation genetics, where many species have reduced genetic variation. As in the koala, due to the recent *phaCin-β* activity, ERV loci by this lineage may show higher diversity than SNPs[28], and could help disentangle genetic relatedness for population conservation management, breeding plans or translocation strategies.

Our study strongly suggests a high ERV burden in koalas, stemming from both the historical expansion of the *phaCin-β* lineage, and the current accumulation of KoRV integrations within QLD and NSW populations. However, the cumulative impact of these ERVs on the fitness of koala individuals remains to be determined. There is indication of *phaCin-β*

expression difference between QLD and SA koalas[29] (note: *phaCin-β* loci were annotated as "syncytin-like"). Due to sequence similarities between *phaCin-β* loci, however, sequencing reads from *phaCin-β* insertions will often mis-map to other *phaCin-β* loci in the reference genome, making it particularly difficult to determine locus specific expression in this ERV lineage. Nevertheless, expression of *phaCin-β* ERVs detected in SA and QLD implies that they may have a functional impact and that further work is required to determine these potential effects.

In conclusion, we present contrasting polymorphism patterns for three recent (in evolutionary terms) ERV lineages across the koala population and reveal thousands of ERVs among the 430 sequenced koala individuals. We show clear population differences for the most recently, possibly still active, *phaCin-β* and KoRV lineages that illuminate recent retrovirus-host interactions and present a resource for population conservation genetics of the endangered koala.

## Methods
### Whole genome sequencing data
Unassembled short-read sequences from 430 koalas (bam files; average coverage: 32.3x, range: 11.3–66.8x) were accessed from the Amazon Web Services Open Data platform (https://registry.opendata.aws/australasian-genomics; accessed 2023/04/11)[12]. Visualization and inspection of candidate loci used the Integrative genomics viewer, IGV[30].

### Polymorphic reference ERV loci
Deletions were detected using Delly v 0.7.7[31] and Lumpy v 0.3.0[32], then compared to the koala candidate ERV regions list that was formed from RetroTector ERVs in R[33] using the package Genomic Ranges[34], regions identified by BLAT with similarity to full length ERVs in the three focal expansions (non-overlapping with RetroTector identified loci), and regions identified by BLAT with similarity to the LTRs of ERVs the three focal expansions (non-overlapping with the two previous datasets). Inaccurate Delly calls for KoRVs were removed after visual inspection. For deletions called by both softwares with comparable lengths, the Delly genotypes were retained. Unique Delly and Lumpy calls were both kept.

### Non-reference ERV insertional loci
Insertions identified by Retroseq[35] as previously described[11]. Briefly, ERV loci were detected by RetroSeq[35] using a custom ERV reference library (-eref), formed from Koala ERVs and retrovirus sequences from diverse lineages. Retroseq *call* using softclips and minimum two reads were used across the samples. Calls were filtered in R to those with FL 8, CLIP3 ≥ 2 and CLIP5 ≥ 2, cov/2 ≤ minGQ ≤ 2 * cov in order to define ERV loci, which were then restricted to the three focal lineages: KoRV, *phaCin-β* and *phaCin-β-like* (using a strict filter that all ERVs called at a locus were all called for the same lineage). During this process, a number of putative Retroseq loci were identified as containing calls for two ERV lineages and were inspected manually. After inspection, 24 such loci that contained two insertions by different lineages separated by only a very small distance were manually curated. Overlaps to reference ERVs were removed unless insertion was identified in another ERV lineage, as this represented secondary insertion. RetroSeq filters were relaxed (FL3; GC2) to count common ERV insertions across individuals.

### Genomic locations of ERV integrations
KoRV, *phaCin-β* and *phaCin-β-like* ERV accumulation in the koala genome was inspected by plotting locus midpoints across the largest scaffolds in the reference assembly. Genomic regions (20 kb windows) with multiple integrations were identified using a binnedSum in R.

### Population comparisons
ERV counts per individual were adjusted to account for differences in coverage (corrcount = ERV count within sample / sample coverage * mean coverage for all samples). Plots were generated using R and ggplot[36]. Heatmaps were generated with the pheatmap package in R.

### Comparisons between reference assemblies
BLAT was used to identify *phaCin-β* loci in the South Australian (SA) koala assembly (Genbank accession: GCA_030178435.1), using both full-length *phaCin-β* and LTR sequences. Flanking regions (1 kb upstream and downstream) of candidate proviruses and solo LTRs were extracted from the SA assembly and lifted over to the reference genome using BLAT. The reverse process then lifted over *phaCin-β* loci from the reference genome to the SA assembly. Comparisons between these two assemblies were manually inspected and summarized.

### Reporting summary
Further information on research design is available in the Nature Portfolio Reporting Summary linked to this article.

## Data availability
The reference koala assembly is available at Genbank: GCA_002099425.1 [https://www.ncbi.nlm.nih.gov/datasets/genome/GCF_002099425.1/]. The South Australian koala assembly is available at Genbank: GCA_030178435.1 [https://www.ncbi.nlm.nih.gov/datasets/genome/GCA_030178435.1/]. Whole genome sequencing data are available from Amazon Web Services Open Data platform (https://registry.opendata.aws/australasian-genomics/)[12].

## Code availability
Code and supporting files are available at GitHub: (https://github.com/PatricJernLab/Koala_ERVs_population_screening; https://doi.org/10.5281/zenodo.10782209).

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

## Acknowledgements

We would like to thank Jason Hill for helpful suggestions and comments. Data was produced as part of the Koala Genome Survey with funding from the NSW Government and the Australian Government's Bushfire Recovery for Wildlife and their Habitats program (GA-2000526), further support was provided by The University of Sydney, Amazon Web Services Open Data Sets, Ramaciotti Centre for Genomics and Illumina. The data was derived from samples provided by Amber Gillett, Amy Shima, Australian Museum, Australian Museum Frozen Tissue Collection, Ben Moore, Carolyn Hogg, Enhua Lee, Fiona Hogan, Karen Marsh, Lachlan Wilmott, Lyndal Husle, Mark Krockenberg, Michaela Blyton, Peter Timms, Rachel Labador, and Taronga Western Plains Zoo. This work was funded by Swedish Research Council VR Grants 2021-01740 (to P.J.) and 2021-04238 (to M.L.) and FORMAS Grant 2018-01008 (to P.J.). The computations/data handling were provided by the Swedish National Infrastructure for Computing (SNIC) at the Uppsala Multidisciplinary Center for Advanced Computational Science, partially funded by the Swedish Research Council through Grant Agreement 2018-0597 (Projects SNIC2022/22-945, NAISS-2023/23-14).

## Author contributions

M.L., M.E.P. and P.J. designed research; M.L. carried out endogenous retrovirus genotyping; M.L., M.E.P. and P.J. analyzed data; M.L. drafted the initial manuscript. M.L. and P.J. wrote the paper.

## Funding

## Competing interests

The authors declare no competing interests.
