## [Peer Review File · Communications Biology]

Reviewers' comments:

Reviewer #1 (Remarks to the Author):

Re: Contrasting segregation patterns among endogenous retroviruses across the koala population.

Summary:

In this manuscript, Lillie et al. utilise whole-genome sequencing of 430 koalas made available through the Koala Genome Survey to analyse the activity of three endogenous retroviruses present within koalas, the well-known koala retrovirus, and phaCin- β and phaCin- β -like which were recently identified by the authors. They present interesting findings, expanding upon those presented in their previous publication, by revealing key geographic differences between the ERV lineages. Whilst KoRV was already known to be more prevalent in QLD/NSW compared to Victoria/SA, the authors draw key comparisons between this ERV and phaCin- β , which they found to have a contrasting geographic distribution. The results are clearly written, and the findings provide insight into ERV evolution and virus-host interactions within koalas.

Comments:

Introduction -

Last line of second paragraph (KoRV has wide-ranging disease outcomes....) needs to be amended to better represent the current status of KoRV and disease. KoRV has only been associated with disease at this stage – no causation has been shown. This sentence should be reworded with less conjecture.

Reference 5 is seemingly in the wrong position and appears to belong in the sentence above referencing KoRV prevalence as this manuscript doesn't mention KoRV disease. This reference should be moved to the appropriate position and potentially updated to a more recent and comprehensive publication that details KoRV prevalence more thoroughly (eg. Blyton et al. PNAS 2022).

Reference 12 (Tarlinton et al. Nature 2006) should be included at the end of paragraph 2 as this was the primary paper identifying KoRV as endogenous.

In the opening sentence of the last paragraph, the authors state that two captive populations are included in the Koala Genome Survey. It would be worth including, for completeness, where these captive institutions reside (at least the state).

Results -

Figure 1a – more detail is required in the figure legend. What do the different coloured dots mean? Does dot size indicate sample size at each location? Can a legend be provided that defines the sample size for a given dot size. Readers would also benefit from the location names being provided for simplicity. This would help with interpreting the results from panel B. I know some are listed in panel B of this figure, but readers may not be familiar with these places and their location within each state. A lettering system with a key may be beneficial.

- I also note that this image has been taken from another publication (Hogg et al. 2023). Whilst it is cited, the authors should ensure that the appropriate permissions have been acquired before reproducing this image in their own work. I recommend the authors generate their own version of this image, which can be done quite easily. This will get around originality issues but will also provide more flexibility for addressing the points I have raised above.

Final sentence of paragraph 1 needs to be reworded for clarity.

The second sentence of the second paragraph referencing LDL receptor sounds like discussion to me. Consider moving this point to the discussion section. A reference should be provided in either case.

The last sentences of paragraphs 3 and 5 and second half of paragraph 4 looks like discussion? This

should be moved accordingly.

The following sentence, "Interestingly, there is a large degree of phaCin- β locus sharing across VIC, which could be the result of founder effect in this population after reintroduction....." should be amended. I assume the authors are referring to the reintroduction of koalas to mainland following a population bottleneck, however, this is already addressed in the discussion section and should not be mentioned here where no context is provided.

I would like to see the final comment of paragraph 3 (Given the stationary lifestyle of koalas.....) elaborated on further (once moved to the discussion section). Whilst the difference in integration sites between the northern and southern states of eastern Australia may be due to varying levels of infection over time, I would like to see some other possibilities discussed as, whilst koalas are mostly stationary animals, they do have interactions with each other and hence opportunities for transmission exist. For example, KoRV is believed to have been introduced in QLD and is spreading south. Could the reverse be the case for phaCin- β ? The longer exposure of the southern animals to the virus could explain their increased number of insertion sites compared to northern animals.

Discussion –

Sentence "Alternatively, given the overlapping age estimates between KoRV and phaCin- β 9, differences in numbers of identified loci...." should read "Alternatively, given the overlapping age estimates between KoRV and phaCin- β 9.....the amount of ERVs that were inherited through the germline.

There are a lot of possible explanations to the data being discussed here but no comparisons or examples from related viruses. I would like to see more references from literature to back up the authors claims.

For example –

Regarding sentence "Since phaCin- β is estimated to have entered the koala before KoRV, but have fewer ERVs with population-wide spread, it is conceivable that infections were less permissive to ERV establishment or possibly suppressed by the koala". I would like to see the authors expand on this last point of koala suppression of ERV integration. Are there known mechanisms by which infections may be less permissive or suppressed by the host? This should be backed up with references.

First sentence of paragraph 4, northern and southern should not be capitalised.

Sentence, "We observe this genetic divide between the northern and southern states in our data, with a marked difference in KoRV integration counts across the regions", should be further explained. Differences in KoRV profile between these states may be due to factors other than the genetic divide.

Be careful with statements such as "VIC has prominent locus sharing across the whole sampled region, which would result from founder effect". There may be alternative explanations here, especially as you have already highlighted genomic hotspots above. Such statements should be reworded with less definitive language. Amend throughout manuscript.

Reviewer #2 (Remarks to the Author):

Lille et al. studied the distribution of about 13,000 proviruses belonging to 3 different groups of ERVs at distinct genomic loci as extracted from a recent database of over 400 koala genomes distributed across Australia. One of the groups is the well-studied gammaretrovirus Koala retrovirus (KoRV; the other 2 (phaCin- β and phaCin- β -like) are betaretroviruses recently identified by this group. Although the distribution of KoRV had been reported previously, and is conformed here, by adding the 2 other groups, and extending the analysis to such a large number of individuals, this work stands to make a

significant contribution to our understanding of the spread and endogenization of retroviruses in this important animal model. That said, I have only one major issue that I think the authors should seriously consider before resubmission. As I read this manuscript for the first time, I kept expecting the analyses based on the use of integration sites to identify orthologous proviruses (referred to as loci) to be accompanied by some analyses (e.g., phylogeny) based on the sequences of the proviruses themselves. I think that such analyses have the potential to greatly enhance the insights into the mechanisms leading to the observed distribution: for example, inferring the relative importance of horizontal spread of virus to host migration, or whether the singlets are directly descended from older proviruses in the same group of animals. This submission, I believe, would be very much stronger with such analyses included. Admittedly trees with 13,000 individual sequences might be a bit challenging to present, but I am sure the authors can find a way to simplify the presentation without losing the essence of the result. Random sampling of individual sequences or compressing terminal branches are strategies that come to mind. LTR trees might be particularly useful, if the phylogenetic signal is adequate. Also, Figures like those in Fig. 2 c, f, and I could be used with the heatmaps indicating levels of similarity between ERV sequences relative to their frequency in the genomes.

Other issues:

P 4, L 13: It's unclear (at least to me) what the authors mean by "frequency" here. I would expect to see a plain number. Frequency of what in what?

L 19: "unevenly" A property of random distributions is that they look uneven. A statistical test for randomness is called for here.

L 7 up: "of which" should be "whose."

L 2-4 up I don't understand the basis for this conclusion. First, by "time" authors mean evolutionary time? If so, the assumption appears to be that accessibility of a genome region has inevitably varied during evolution. I doubt very much that this is always true. Possibly I am misunderstanding, but in any case, the authors need to clarify (or remove). This conclusion.

L 2 up-P3, L 1:

P 3, L 6: "...prevalence' and KoRV distribution."

P 7, para 2, L8. This could also relate to time since extinction of the *phiCin-β* group.

L4 up: of, not "by."

P8, para 1: Anglicized versions of taxonomic nomenclature should not be italicized (Lentiviral, gammaretroviruses)

L 8-11: As modified by selection against deleterious insertions.

P 8, para 2, L 6 and 8: observed.

L 4 up ERVs, not "they"

L3 up: cases

P 9, para 2, L 9: implies

Para 3, L 1: "three" what? I think "recent" should be here.

P 10, para 2, L 5: were, not "was."

P 11, L 2 up data are plural.

Figure 1 Legend: What does the size of the circles indicate?

Panel A: the colors of the circles don't match the legend.

Figure 2 panels c,f,i: Very hard to see. Please use a different more contrasty palette, with low values in lighter color.

Reviewer #1 (Remarks to the Author):

Re: Contrasting segregation patterns among endogenous retroviruses across the koala population.

Summary:

In this manuscript, Lillie et al. utilise whole-genome sequencing of 430 koalas made available through the Koala Genome Survey to analyse the activity of three endogenous retroviruses present within koalas, the well-known koala retrovirus, and phaCin- β and phaCin- β -like which were recently identified by the authors. They present interesting findings, expanding upon those presented in their previous publication, by revealing key geographic differences between the ERV lineages. Whilst KoRV was already known to be more prevalent in QLD/NSW compared to Victoria/SA, the authors draw key comparisons between this ERV and phaCin- β , which they found to have a contrasting geographic distribution. The results are clearly written, and the findings provide insight into ERV evolution and virus-host interactions within koalas.

We thank the reviewer for positive comments and suggestions.

Comments:

Introduction -

Last line of second paragraph (KoRV has wide-ranging disease outcomes....) needs to be amended to better represent the current status of KoRV and disease. KoRV has only been associated with disease at this stage – no causation has been shown. This sentence should be reworded with less conjecture.

We reworded the sentence:

“KoRV has been associated with wide-ranging disease outcomes”

Reference 5 is seemingly in the wrong position and appears to belong in the sentence above referencing KoRV prevalence as this manuscript doesn't mention KoRV disease. This reference should be moved to the appropriate position and potentially updated to a more recent and comprehensive publication that details KoRV prevalence more thoroughly (eg. Blyton et al. PNAS 2022).

The reference (Simmons et al 2012) was moved to the preceding sentence and we added Blyton et al PNAS 2022 here as well.

Reference 12 (Tarlinton et al. Nature 2006) should be included at the end of paragraph 2 as this was the primary paper identifying KoRV as endogenous.

The reference (Tarlinton et al 2006) was added to the last sentence of paragraph 2.

In the opening sentence of the last paragraph, the authors state that two captive populations are included in the Koala Genome Survey. It would be worth including, for completeness, where these captive institutions reside (at least the state).

Information about the two captive populations was added

Results -

Figure 1a – more detail is required in the figure legend. What do the different coloured dots mean? Does dot size indicate sample size at each location? Can a legend be provided that defines the sample size for a given dot size. Readers would also benefit from the location names being

provided for simplicity. This would help with interpreting the results from panel B. I know some are listed in panel B of this figure, but readers may not be familiar with these places and their location within each state. A lettering system with a key may be beneficial.

- I also note that this image has been taken from another publication (Hogg et al. 2023). Whilst it is cited, the authors should ensure that the appropriate permissions have been acquired before reproducing this image in their own work. I recommend the authors generate their own version of this image, which can be done quite easily. This will get around originality issues but will also provide more flexibility for addressing the points I have raised above.

We generated a new map image and labelled sampling locations according to the numbered locations in Fig. 1b, and kept the reference to Hogg et al. 2023 in the figure legend.

Final sentence of paragraph 1 needs to be reworded for clarity.

We reworded this sentence for improved clarity:

“There were still relatively many rare *phaCin-β-like* loci, with 151 (32.2 %) *phaCin-β-like* insertional polymorphisms found in only one individual.”

The second sentence of the second paragraph referencing LDL receptor sounds like discussion to me. Consider moving this point to the discussion section. A reference should be provided in either case.

We included a new reference (Príncipe et al 2021)

The last sentences of paragraphs 3 and 5 and second half of paragraph 4 looks like discussion? This should be moved accordingly.

We moved the sentences from paragraph 3 and 5 to the discussion paragraph 4 and expanded the text.

The following sentence, “Interestingly, there is a large degree of *phaCin-β* locus sharing across VIC, which could be the result of founder effect in this population after reintroduction.....” should be amended. I assume the authors are referring to the reintroduction of koalas to mainland following a population bottleneck, however, this is already addressed in the discussion section and should not be mentioned here where no context is provided.

We have amended this sentence:

“There is a large degree of *phaCin-β* locus sharing across VIC with intermediate frequencies of many *phaCin-β* ERV loci private to the state, which may reflect founder effect...”

I would like to see the final comment of paragraph 3 (Given the stationary lifestyle of koalas.....) elaborated on further (once moved to the discussion section). Whilst the difference in integration sites between the northern and southern states of eastern Australia may be due to varying levels of infection over time, I would like to see some other possibilities discussed as, whilst koalas are mostly stationary animals, they do have interactions with each other and hence opportunities for transmission exist. For example, KoRV is believed to have been introduced in QLD and is spreading south. Could the reverse be the case for *phaCin-β*? The longer exposure of the southern animals to the virus could explain their increased number of insertion sites compared to northern animals.

This sentence was moved to discussion paragraph 4 and elaborated:

“We also observed differences in the pattern of *phaCin-β* integrations across the states. Given the stationary lifestyle of koalas, these patterns could reflect the historical intensity of infection, which likely differ between the two retrovirus lineages. Comparisons between the provirus content of the SA genome relative to the northern Bilbo genome would imply more recent *phaCin-β* activity in the southern koala populations. The patterns in VIC, though, appear to be shaped by a founder effect (Supplementary Fig 3). VIC has prominent locus sharing across the whole sampled region, in contrast to the patterns in QLD and NSW, which could be the result of a founder effect, after populations were reintroduced from island refugia. The relative contributions from any founder effect on ERV loci, compared to the recent, potentially ongoing, *phaCin-β* activity in these southern population will require further study. In particular, investigating the proviral content and sequence divergence of the ERV integrations in VIC populations, for example using long-reads, will greatly inform our understanding of the two processes.”

With regard to directionality of the *phaCin-β* spread, it is rather difficult to speculate at this time, given the limited available information regarding complete ERV locus content. Future studies, making use of long read technology will likely provide better information about the entire ERV sequences (i.e. full-length provirus from recent integration, older proviruses or solo LTRs), which can be used to estimate the age of integrations for relevant conclusions about the history of *phaCin-β* expansion in the koala population. At this stage, based on the two reference assemblies where we can compare integrations, ERV content and provirus sequences, we are only in a position to suggest there has been more recent activity in the south.

Discussion –

Sentence “Alternatively, given the overlapping age estimates between KoRV and *phaCin-β9*, differences in numbers of identified loci....” should read “Alternatively, given the overlapping age estimates between KoRV and *phaCin-β9*.....the amount of ERVs that were inherited through the germline.

This has been corrected

There are a lot of possible explanations to the data being discussed here but no comparisons or examples from related viruses. I would like to see more references from literature to back up the authors claims.

For example –

Regarding sentence “Since *phaCin-β* is estimated to have entered the koala before KoRV, but have fewer ERVs with population-wide spread, it is conceivable that infections were less permissive to ERV establishment or possibly suppressed by the koala”. I would like to see the authors expand on this last point of koala suppression of ERV integration. Are there known mechanisms by which infections may be less permissive or suppressed by the host? This should be backed up with references.

We elaborated on this with additional references in the text:

“A host of mechanisms against retroviral replication have been described^{1,2}, but remain uncharacterized in koala. Such an innate defense mechanism in koala could involve Piwi-interacting RNAs (piRNAs), which normally suppress transposition *in trans* by means of anti-sense

transcripts but appears to inhibit KoRV replication *in cis* by sense strand piRNA from unspliced KoRV transcripts that halt replication²⁰.”

First sentence of paragraph 4, northern and southern should not be capitalised.

This has been corrected

Sentence, “We observe this genetic divide between the northern and southern states in our data, with a marked difference in KoRV integration counts across the regions”, should be further explained. Differences in KoRV profile between these states may be due to factors other than the genetic divide.

Added: “..., reflecting the geographical differences in KoRV prevalence.”

Be careful with statements such as “VIC has prominent locus sharing across the whole sampled region, which would result from founder effect”. There may be alternative explanations here, especially as you have already highlighted genomic hotspots above. Such statements should be reworded with less definitive language. Amend throughout manuscript.

We amended sentence to “which could be the result of founder effect...”

Reviewer #2 (Remarks to the Author):

Lille et al. studied the distribution of about 13,000 proviruses belonging to 3 different groups of ERVs at distinct genomic loci as extracted from a recent database of over 400 koala genomes distributed across Australia. One of the groups is the well-studied gammaretrovirus Koala retrovirus (KoRV; the other 2 (phaCin- β and phaCin- β -like) are betaretroviruses recently identified by this group. Although the distribution of KoRV had been reported previously, and is conformed here, by adding the 2 other groups, and extending the analysis to such a large number of individuals, this work stands to make a significant contribution to our understanding of the spread and endogenization of retroviruses in this important animal model. That said, I have only one major issue that I think the authors should seriously consider before resubmission. As I read this manuscript for the first time, I kept expecting the analyses based on the use of integration sites to identify orthologous proviruses (referred to as loci) to be accompanied by some analyses (e.g., phylogeny) based on the sequences of the proviruses themselves. I think that such analyses have the potential to greatly enhance the insights into the mechanisms leading to the observed distribution: for example, inferring the relative importance of horizontal spread of virus to host migration, or whether the singlets are directly descended from older proviruses in the same group of animals. This submission, I believe, would be very much stronger with such analyses included. Admittedly trees with 13,000 individual sequences might be a bit challenging to present, but I am sure the authors can find a way to simplify the presentation without losing the essence of the result. Random sampling of individual sequences or compressing terminal branches are strategies that come to mind. LTR trees might be particularly useful, if the phylogenetic signal is adequate. Also, Figures like those in Fig.2 c, f, and I could be used with the heatmaps indicating levels of similarity between ERV sequences relative to their frequency in the genomes.

We thank the reviewer for positive comments and suggestions.

We agree that additional phylogenetic analyses and characterizing the provirus sequences themselves would provide valuable and important insights into the infection and transmission across the population, and to date integrations. However, with the available short-read sequencing data, it is impossible to re-construct these provirus sequences from the integration sites that we have identified. Short-reads containing ERV sequence in the sequenced individuals rely on chromosomal anchoring for locus identification. Without this anchoring, short reads typically mismap to the, abundant, similar ERVs found in the reference assembly, as they cannot all map to their true location in the individual's genome. The only information we have for integrations is the signatures of insertions, where one paired read maps to the reference assembly coordinates, and the other to the ERV sequence (primarily in the LTR and not much further into the provirus), and the characteristic chromosomal target site duplications (TSDs) flanking the proviral insertion.

We recently performed phylogenetic analysis of ERVs the reference assembly (Lillie et al PNAS 2022), where we characterised *phaCin-β* and *phaCin-β*-like elements and presented their evolutionary relationships with other retroviruses, and here we also reflected on the very shallow divergence with KoRV and *phaCin-β* clades, and occurrence of identical proviruses in the reference assembly.

We anticipate that technological developments and reduced costs associated with long-read sequencing will overcome some of the hurdles, as whole genome sequencing with long reads brings with the prospect of being able to directly characterize the ERV content (whether it be solo LTR, full-length provirus, or aberrated provirus due to secondary integrations/deletions/mutations) within individuals and to be able to make significant conclusions about infection transmission, integration dates, and a better understanding about the fate of ERVs in the genome.

Other issues:

P 4, L 13: It's unclear (at least to me) what the authors mean by "frequency" here. I would expect to see a plain number. Frequency of what in what?

We have clarified what we mean by frequency:

"The timing of retroviral activity was also reflected in the frequency distribution of ERV insertional polymorphisms across the population."

We also changed the use of "frequency" to "number" on page 7 to avoid confusion.

L 19: "unevenly" A property of random distributions is that they look uneven. A statistical test for randomness is called for here.

We included a binomial test and updated the text:

"Assuming a neutral model of random integrations where the success rate equals the number of unique ERVs divided by the genome length, we would not expect to find any 20kb window with 5 or more ERVs (exact binomial test $p \approx 8.3 \times 10^{-8}$). However, we observe 166 such windows with 5 or more integrations. We also observe potential "coldspot" regions of the genome, such as the 3.7 Mb region MST01000019.1:5,903,013-9,671,599 where no integrations

by the three focal ERV lineages KoRV, *phaCin-β*, and *phaCin-β-like* were detected in our population-wide screening. Assuming the same neutral model (above), the probability of observing a gap of this length can be estimated to $p \approx 2.9 \times 10^{-7}$.”

L 7 up: “of which” should be “whose.”

This has been corrected

L 2-4 up I don’t understand the basis for this conclusion. First, by “time” authors mean evolutionary time? If so, the assumption appears to be that accessibility of a genome region has inevitably varies during evolution. I doubt very much that this is always true. Possibly I am misunderstanding, but in any case, the authors need to clarify (or remove). This conclusion.

We removed this speculation

L 2 up-P3, L 1:

See above

P 3, L 6: “...prevalence’ and KoRV distribution.”

Added:

“...and KoRV distribution”

P 7, para 2, L8. This could also relate to time since extinction of the *phaCin-b* group.

Added:

“..or time since the potential extinction of *phaCin-β* (although we cannot exclude *phaCin-β* from still being active)”

L4 up: of, not “by.”

This has been corrected

P8, para 1: Anglicized versions of taxonomic nomenclature should not be italicized (Lentiviral, gammaretroviruses)

This has been corrected

L 8-11: As modified by selection against deleterious insertions.

Added:

“..selection against deleterious insertions in particular regions of the genome, or..”

P 8, para 2, L 6 and 8: observed.

This has been corrected

L 4 up ERVs, not “they”

This has been corrected

L3 up: cases

This has been corrected

P 9, para 2, L 9: implies

This has been corrected

Para 3, L 1: “three” what? I think “recent: should be here.

This sentence has been restructured for clarity

P 10, para 2, L 5: were, not “was.”

This has been corrected

P 11, L 2 up data are plural.

This has been corrected

Figure 1 Legend: What does the size of the circles indicate?

We generated a new map and included sampling locations according to numbers and location naming in Fig. 1b.

Panel A: the colors of the circles don't match the legend.

We generated a new map and included sampling locations according to numbers and location naming in Fig. 1b. We refer to the sampling in the original publication (Hogg et al. 2023) in the legend.

Figure 2 panels c,f,i: Very hard to see. Please use a different more contrasty palette, with low values in lighter color.

We have regenerated heatmaps to increase contrast.

REVIEWERS' COMMENTS:

Reviewer #1 (Remarks to the Author):

Re: Contrasting segregation patterns among endogenous retroviruses 1 across the koala population

The authors have adequately addressed my comments and I endorse the publication of this version of the manuscript.

Reviewer #2 (Remarks to the Author):

The authors have effectively responded to most of the points raised in my review, and I would consider this manuscript suitable for publication once they deal with one easily fixed issue. In my previous review, the major concern was the lack of phylogenetic analysis to illuminate the relationship and timing of the impressive number of proviruses identified in this study. In response, the authors pointed out that the sequence data used were all short-read, and, although proviruses could be reliably identified by their virus-host junctions, it was not possible to assemble their internal sequences without longer reads. I accept that explanation but note that other readers are very likely to be thinking the same thing as they read the paper. On reading this version, I realized that the nature of the underlying data were not at all clear, except for one rather elliptical sentence (lines 221-223) near the end of the Discussion. To make the reader better informed as to just what was done, I suggest the following:

1. At the end of line 54, insert "unassembled."
2. Line 70: After "loci" add something like "as characterized by unique ERV-host junctions."
3. In the first section of Methods, make it clear that the data are unaligned short reads, particularly since "alignment files" implies the opposite.

A couple of other small points:

Line 119 "loci...Have occurred" sounds weird. "germline integrations" or some such would be better here.

Line 161: which, not "that."

With these few changes, I think the manuscript is good to go.

Reviewer #1 (Remarks to the Author):

The authors have adequately addressed my comments and I endorse the publication of this version of the manuscript.

We thank the reviewer for this positive comment.

Reviewer #2 (Remarks to the Author):

The authors have effectively responded to most of the points raised in my review, and I would consider this manuscript suitable for publication once they deal with one easily fixed issue.

In my previous review, the major concern was the lack of phylogenetic analysis to illuminate the relationship and timing of the impressive number of proviruses identified in this study. In response, the authors pointed out that the sequence data used were all short-read, and, although proviruses could be reliably identified by their virus-host junctions, it was not possible to assemble their internal sequences without longer reads. I accept that explanation but note that other readers are very likely to be thinking the same thing as they read the paper. On reading this version, I realized that the nature of the underlying data were not at all clear, except for one rather elliptical sentence (lines 221-223) near the end of the Discussion. To make the reader better informed as to just what was done, I suggest the following:

We thank the reviewer for these additional suggestions.

1. At the end of line 54, insert “unassembled.”

Done.

2. Line 70: After “loci” add something like “as characterized by unique ERV-host junctions.”

Done.

3. In the first section of Methods, make it clear that the data are unaligned short reads, particularly since “alignment files” implies the opposite..

Done.

A couple of other small points:

Line 119 “loci...Have occurred” sounds weird. “germline integrations” or some such would be better here.

Done.

Line 161: which, not “that.”

Done.

With these few changes, I think the manuscript is good to go.